# Change of Human Footprint in China and Its Implications for Carbon Dioxide (CO$_2$) Emissions

Yuan Li [1,2], Wujuan Mi [2], Yuheng Zhang [2], Li Ji [1], Qiusheng He [3], Yuanzhu Wang [4] and Yonghong Bi [2,*]

1 School of Environment and Resources, Taiyuan University of Science and Technology, Taiyuan 030024, China
2 State Key Laboratory of Freshwater Ecology and Biotechnology, Institute of Hydrobiology, Chinese Academy of Sciences, Wuhan 430072, China
3 Institute of Intelligent Low Carbon and Control Technology, Taiyuan University of Science and Technology, Taiyuan 030024, China
4 Central Southern Safety & Environmental Technology Institute Company Limited, Wuhan 430072, China
* Correspondence: biyh@ihb.ac.cn; Tel.: +86-27-68780016

**Abstract:** Humans have altered the earth in unprecedented ways, and these changes have profound implications for global climate change. However, the impacts of human pressures on carbon dioxide (CO$_2$) emissions over long time scales have not yet been clarified. Here, we used the human footprint index (HF), which estimates the ecological footprint of humans in a given location, to explore the impacts of human pressures on CO$_2$ emissions in China from 2000 to 2017. Human pressures (+13.6%) and CO$_2$ emissions (+198.3%) in China are still on the rise during 2000–2017 and are unevenly distributed spatially. There was a significant positive correlation between human pressures and CO$_2$ emissions in China, and northern China is the main driver of this correlation. The increase of CO$_2$ emissions in China slowed down after 2011. Although human pressures on the environment are increasing, high-quality development measures have already had noticeable effects on CO$_2$ emission reductions.

**Keywords:** carbon dioxide emissions; global climate change; human footprint index; human pressures; macro control

## 1. Introduction

Humans have been reshaping the world for millennia. The current geological epoch has been renamed the "Anthropocene" [1] because of the major environmental impacts associated with human activities, especially the conversion of large-scale natural habitats to cropland and the construction of land to feed and house our burgeoning population [2]. Human activities affect the stability of ecosystems and have a multitude of deleterious effects on the environment, including nutrient pollution [3]; modifications of land surface hydrology [4]; loss of the productivity, composition, and diversity of terrestrial ecosystems [5]; introduction of alien species; and alterations of the biogeochemical cycles of carbon [6]. Scientists have expressed much concern over the extent of human-induced environmental destruction, which is rapidly approaching catastrophic levels.

As a consequence of the cumulative impacts of anthropogenic activities on the planet in recent decades, the concentrations of greenhouse gases including carbon dioxide (CO$_2$), methane (CH$_4$), and nitrous oxide (N$_2$O), have been continuously increasing [7]. The accumulation of greenhouse gases prevents the loss of heat from the earth and increases the temperature of the earth's surface, which results in an increase in the frequency and intensity of extreme weather events, including drought, flood, heat waves, and freezing stress [8,9]. The 2015 Paris Agreement ambitiously aims to limit warming to between 1.5 °C and 2 °C by mid-century, which will require achieving a balance between anthropogenic emissions and the removal of longer-lived greenhouse gases such as CO$_2$ [10]. Many

countries, such as China and the United States, have made net-zero pledges and have started to reduce emissions [11].

The extent and intensity of anthropogenic changes are spatially heterogeneous. Many anthropogenic pressures, such as population growth, fossil fuel combustion, industrial emissions, agricultural production, livestock farming, and land-use change, can interact in diverse ways [12,13]. Currently used mapping approaches often fail to capture many lower-intensity forms of human pressures, such as our extensive networks of roads, grazing lands, and low-density human settlements, the effects of which are more insidious than outright habitat conversion [14]. For example, land-use cover has major effects on the global extent and distribution of terrestrial carbon emissions [15]. Changes in land-use type from high-vegetation to low-vegetation biomass usually result in carbon emissions; land-use management, such as measures to control wildfires, pests, and diseases, can also affect carbon storage. Land-use change and management have together been estimated to contribute approximately one-third of all anthropogenic carbon emissions since the industrial revolution [16,17]. Non-settlement areas can also potentially result in population displacement and enrichment [2]; however, the role of low stress factors on the environment is often ignored in such areas. A comprehensive human stress indicator of multiple factors for studying the spatiotemporal patterns of $CO_2$ emissions is often more effective for characterizing regions with high emissions [18]. Furthermore, key variables, including technology, infrastructure, environment and finance, have important practical significance in achieving carbon neutrality [19–21]. For example, Labzovskii et al., (2017) [22] have projected that a beneficial policy would result in 24%, 80%, 166% less $CO_2$ emissions in East Asia by 2020, 2025 and 2030, respectively.

Improvements in remote sensing geographic data and geographic information systems permit the construction of detailed maps displaying human activities, such as the human footprint index (HF) [23]. HF captures the total ecological footprint of the human population in a given location [24]. Venter et al., constructed a global human footprint map from 1993 to 2009 and found that intense human pressures have had significant effects on native biodiversity [25]. This study also revealed that environmental pressures were lower in the wealthiest countries. A series of studies have also shown that human activities have had substantial effects on marine and lake ecosystems through the construction of human footprint maps [26,27]. However, the macro-scale effects of human pressures on $CO_2$ emissions have not yet been thoroughly studied.

China has experienced rapid economic development since its reform and opening in the late 1970s, which has resulted in the emission of large amounts of $CO_2$. A better understanding of spatiotemporal patterns in human pressures and their effects on $CO_2$ emissions is needed to respond to calls for rapid action to limit $CO_2$ emissions. Here, we used the latest multi-source remote sensing data to quantify the effects of human pressures on China's regional and sectoral $CO_2$ emissions between 2000 and 2017. Remote sensing data on (1) land-use cover, (2) roads, (3) railways, (4) human population density, (5) grazing density, and (6) night-time lights were used. We explored the spatiotemporal dynamics in human pressures and $CO_2$ fluxes and analyzed the correlations between human pressures and $CO_2$ fluxes. Our results reveal that $CO_2$ emissions in China have been on the rise from 2000 to 2017 and that there is a strong correlation between $CO_2$ emissions and human pressures. Nevertheless, increases in $CO_2$ emissions have slowed down after 2011, indicating that the effects of high-quality development and national measures to reduce greenhouse gas emissions have started to have impacts on $CO_2$ emission reductions.

Our results enhance the understanding of the impacts of human pressures on the environment and have implications for the formulation of effective and environmentally friendly strategies. The results have a positive guiding significance for carbon emission reduction policies, including urban land planning, population size control, industrial structure optimization, industrial technology upgrading, etc. In this paper, Section 2 provides a brief overview of the study methods, data sources, and model construction. Then, the spatio-temporal characteristics of HF and $CO_2$, and their correlation analysis

during 2000–2017 in China are provided in Section 3. The cause of spatial distribution of HF and $CO_2$, and the response and driving factors of $CO_2$ emissions to human pressure changes are discussed in Section 4. Finally, Section 5 outlines the potential extension, outlook, and the summary of this study.

## 2. Methods and Data

### 2.1. Overview

The human footprint (HF) is a global map of human influence on the land surface [28]. The human footprint can reflect human disturbance to the natural environment and is directly or indirectly related to human fossil fuel combustion, fertilizer use, and industrial activity, thus establishing a certain relationship with CO2 emissions [29]. We used the HF for data from 2000 to 2020 to facilitate comparison across human pressures. The human pressures considered were (1) land-use cover, (2) roads, (3) railways, (4) human population density, (5) night-time lights, and (6) grazing density. We performed buffer analysis and assigned scores ranging from 0 to 10 to various spatial data layers according to the intensity of each human pressure; the assigned layers were then overlaid and normalized by partitioning to obtain human footprint data that indicate the extent of terrestrial human activity. We used ArcGIS 10.2 to integrate spatial data on human pressures at the 1 km$^2$ resolution for China's land areas. For any grid cell, the human footprint ranged between 0 and 60. We classified HF into three categories to reflect the degree of human pressures: low (HF < 20), moderate (HF 20–30), and high (HF 30–60).

### 2.2. Land-Use Cover

Change in land-use cover reflects the long-term impact of human activities on the environment [30]. Anthropogenic land-use cover change has a pronounced effect on regional climate change [31]. For example, land-use change (e.g., deforestation, increases in cropland, livestock rearing, and urban and industrial land expansion) and land management (wildfires, pests, and diseases) can directly or indirectly increase $CO_2$ emissions, especially in developing and poor countries [32–35]. Overall land-use change and land management have contributed approximately 1.45 Pg of the total carbon released from 1990 to 2010 [36]. We downloaded the database files of land-use cover (accurate to 1 km) from the Resource and Environment Science and Data Center (http://www.resdc.cn/, accessed on 1 January 2022). We assigned a pressure score of 10 to construction land, 7 to agricultural land, 4 to grasslands, and 0 to all other types of land-use cover. We used the land-use cover data of 2000, 2005, 2010, 2015, and 2020 to approximate the data for 2000–2003, 2004–2007, 2008–2011, 2012–2015, and 2016–2017, respectively.

### 2.3. Roads and Railways

Transport infrastructure is a fundamental physical foundation of societies; it thus plays a key role in supporting socio-economic activities and affects both the local and global environment [37,38]. Road transport contributes more than 60% of the $CO_2$ emissions of all transport activities [39], and a unidirectional causality has been noted between railway infrastructure and energy consumption [40]. An increasing number of transport infrastructure projects have been implemented in newly urbanizing regions.

We acquired data on the distribution of roads and railways for the years 2000, 2005, 2010, and 2012–2017 from OpenStreetMap. The OpenStreetMap database represents the most comprehensive publicly available database on roads and railways. We excluded all trails and private roads and used provincial major highways to denote roads. We used the road network data of 2000, 2005, and 2010 to approximate the data for 2000–2003, 2004–2007, and 2008–2011, respectively. We evaluated the direct and indirect pressures of roads by designating a pressure score of 10, 8, and 4 for 0.5, 0.5–1.5, and 1.5–2.5 km of distance on either side of the roads. The direct pressure of railways was assigned a score of 8 for a distance of 0.5 km on either side of them.

### 2.4. Population Density

Population density is an important indicator of the intensity of the interaction between human activities and ecosystems, and there is a strong correlation between population density and $CO_2$ emissions. Population density has significant positive effects on increases in $CO_2$ emissions. A previous study has shown that population density increases of 1% lead to total net $CO_2$ emissions increases of 4% [41,42].

Population density was mapped using the gridded population data published by WorldPop. The data set provides a 1 km × 1 km gridded summary of population census data from 1990 to 2017. Random forest-based asymmetric redistribution was used for mapping. The impact of population density on ecosystems is logarithmic. The population density data were assigned a pressure score ranging from 0 to 10. The maximum value of population density was 366,587 persons/km$^2$, and the population density pressure score was logarithmically scaled using the following formula:

$$\text{Population score} = 2.21398 \times \log (\text{Population density} + 1) \tag{1}$$

### 2.5. Grazing Density

Overgrazing (i.e., heavy grazing) associated with the rapid sharp growth of the human population and food demand in recent years is a major contributor to increases in greenhouse gas fluxes [43].

We tallied the number of large livestock animals, including cattle, horses, donkeys, mules, and camels, kept in each provincial administrative region of China using information published by the National Bureau of Statistics of China from 2000 to 2017. The impact of grazing density on ecosystems is logarithmic. The grazing density data were assigned a pressure score of 0–10. The maximum value of grazing density was 93.327 heads/km$^2$ (calculated by provincial administrative unit), and the grazing density pressure score was logarithmically scaled using the following formula:

$$\text{Livestock score} = 2.51531 \times \log (\text{grazing density} + 1) \tag{2}$$

### 2.6. Night-Time Lights

Night-time light satellite imagery is correlated with socioeconomic parameters such as urbanization, economic activity, and population density [44–46]. Night-time light imagery of 2000–2013 was obtained from the Defense Meteorological Satellite Program–Operational Linescan System (DMSP–OLS). The night-time light imagery of 2014–2020 was obtained from the launch of the NASA/NOAA's Suomi—Visible Infrared Imaging Radiometer Suite (VIIRS) sensor.

We used the annual mean synthesis algorithm without the effect of moonlight and cloud cover, and the spatial resolution of the images was 1 km × 1 km. Because of the lack of in-orbit radiometric calibration and correction facilities, the nocturnal radiometric signals on all light images were discretized into digitized radiometric brightness values (hereafter referred to as DN values, with scores ranging from 0 to 63). The synthetic product of this series of stable night-light signals eliminates the effect of short-time radiation sources so that the parts covered by the high brightness DN values usually correspond to high-density human settlements and activities. DN values of 8 or less were excluded from consideration before inter-calibration of data, as the shape of the quadratic function leads to severe distortion of very low DN values. The inter-calibrated DN data were then rescaled using an equal quintile approach into a 0–10 scale.

### 2.7. $CO_2$ Fluxes

The emission inventory of $CO_2$ fluxes was obtained from the latest energy data revision (2015) of the China National Bureau of Statistics from Scientific Data (https://doi.org/10.6084/m9.figshare.c.5136302.v2, accessed on 1 January 2022) [47,48]. The carbon emission unit is accurate to the county administrative region. We collected data on the $CO_2$ emissions

of 2689 county-level administrative regions in China from 2000 to 2017 (Figure S1). The $CO_2$ emission fluxes were classified into three categories: low (0–10 Mt), moderate (10–30 Mt), and high (30–60 Mt).

### 2.8. Statistical Analyses

We performed all simple linear regression analysis, including the determination of 95% confidence intervals, with SigmaPlot 14.0 (Systat Software, San Jose, CA, USA). $r^2$ indicates the fit of a one-dimensional linear regression, and $p < 0.0001$ was the threshold for statistical significance. Standard deviations of $CO_2$ emissions and HF by different regions were calculated using SigmaPlot 14.0 software. We evaluated bivariate comparisons of continuous data measurements using analysis of variance (ANOVA) tests. $CO_2$ emissions and HF by different regions were expressed as average and standard error (SE). We tested the errors of HF calculation using a Monte Carlo simulation with 1000 iterations. We used histograms, conducted linear regressions, and built line plots. We used Origin Pro 2021 (Microcal Software, Seattle, WA, USA) to make violin plots and stacked area plots. All the spatial patterns of $CO_2$ emissions and HF in China were analyzed by ArcGIS 10.2 (ESRI, Redlands, CA, USA) [49].

## 3. Results

### 3.1. Spatio-Temporal Pattern of Human Footprint in China

HF in China was mainly concentrated in the 0–20 range, spanning $482 \times 10^4$ km$^2$ (Figure 1a). The area of high (30–60), moderate (20–30), and low HF (0–20) in China was $5.53 \times 10^4$ km$^2$, $100.11 \times 10^4$ km$^2$, and $684.09 \times 10^4$ km$^2$, respectively (Table S1). There was high spatial variation in HF (Figure 2a; Figure S2). The largest and smallest areas of high HF were observed in eastern ($1.57 \times 10^4$ km$^2$) and northwestern China ($0.22 \times 10^4$ km$^2$) during 2000–2017. The mean HF in China was 20.89 and gradually increased from 19.84 in 2000 to 22.54 in 2017 (Table 1; Figure 1b). HF in China increased 13.6% during 2000–2017. The rate of change of HF was the highest (16.87%) and the lowest (9.63%) in southern and northern China, respectively (Table 2). The area of low HF was $699.50 \times 10^4$ in 2000 and $645.05 \times 10^4$ km$^2$ in 2017 (a decrease of 7.78%), and the area of high HF was $2.66 \times 10^4$ in 2000 and $12.84 \times 10^4$ in 2017 (an increase of 382.71%) (Table S1). The area of high HF was $0.07 \times 10^4$ in 2000 and $0.99 \times 10^4$ km$^2$ in 2017 (an increase of 1314.29%,) and the area of high HF was $0.67 \times 10^4$ in 2000 and $1.97 \times 10^4$ km$^2$ in 2017 (an increase of 194.03%). Some of the low HF areas were converted into high HF areas (Table S1; Figure 3).

**Table 1.** Average human footprint index in different regions of China during 2000–2017.

| Region/Human Footprint | Nationwide | North | South | Northeast | Northwest | Southwest | Central | East |
|---|---|---|---|---|---|---|---|---|
| 2000 | 19.84 | 22.22 | 17.75 | 20.04 | 15.60 | 17.51 | 21.10 | 22.96 |
| 2001 | 20.30 | 22.62 | 18.25 | 20.50 | 15.86 | 17.97 | 21.60 | 23.56 |
| 2002 | 20.39 | 22.71 | 18.32 | 20.60 | 15.89 | 18.07 | 21.72 | 23.62 |
| 2003 | 20.30 | 22.57 | 18.23 | 20.52 | 15.86 | 18.01 | 21.57 | 23.58 |
| 2004 | 20.66 | 22.99 | 18.72 | 20.89 | 16.07 | 18.16 | 21.82 | 24.22 |
| 2005 | 20.51 | 22.91 | 18.48 | 20.79 | 16.08 | 18.12 | 21.74 | 23.76 |
| 2006 | 20.59 | 22.94 | 18.56 | 20.73 | 16.14 | 18.14 | 21.78 | 24.09 |
| 2007 | 20.62 | 22.89 | 18.61 | 20.84 | 16.14 | 18.20 | 21.82 | 24.07 |
| 2008 | 21.00 | 23.18 | 19.53 | 21.20 | 14.68 | 17.26 | 22.32 | 26.65 |
| 2009 | 20.36 | 22.45 | 18.15 | 21.52 | 15.94 | 18.03 | 21.40 | 23.58 |
| 2010 | 20.80 | 22.83 | 18.58 | 21.82 | 16.20 | 18.28 | 21.93 | 24.41 |
| 2011 | 20.62 | 22.58 | 18.56 | 21.53 | 16.14 | 18.13 | 21.77 | 24.13 |
| 2012 | 20.66 | 22.63 | 18.62 | 21.55 | 16.15 | 18.11 | 21.77 | 24.28 |
| 2013 | 21.01 | 23.01 | 19.08 | 21.82 | 16.42 | 18.26 | 22.13 | 24.80 |
| 2014 | 21.76 | 23.56 | 19.89 | 22.44 | 17.17 | 19.18 | 22.92 | 25.58 |
| 2015 | 21.94 | 23.75 | 20.05 | 22.45 | 17.39 | 19.47 | 23.10 | 25.73 |
| 2016 | 22.06 | 23.86 | 20.23 | 22.40 | 17.47 | 19.60 | 23.28 | 25.84 |

**Table 1.** *Cont.*

| Region/Human Footprint | Nationwide | North | South | Northeast | Northwest | Southwest | Central | East |
|---|---|---|---|---|---|---|---|---|
| 2017 | 22.54 | 24.36 | 20.75 | 22.61 | 17.95 | 20.17 | 23.78 | 26.38 |
| Mean | 20.89 | 23.00 | 18.91 | 21.35 | 16.29 | 18.37 | 22.08 | 24.51 |
| Increment | 2.70 | 2.14 | 3.00 | 2.56 | 2.36 | 2.67 | 2.68 | 3.43 |
| Chang rate (%) | 13.60 | 9.63 | 16.87 | 12.79 | 15.10 | 15.22 | 12.70 | 14.93 |

**Table 2.** Average $CO_2$ emissions in different regions of China during 2000–2017.

| Region/$CO_2$ (Mt) | Nationwide | North | South | Northeast | Northwest | Southwest | Central | East |
|---|---|---|---|---|---|---|---|---|
| 2000 | 3194.81 | 667.21 | 356.84 | 400.53 | 229.78 | 330.06 | 476.33 | 734.06 |
| 2001 | 3217.34 | 669.42 | 350.19 | 386.20 | 226.75 | 319.82 | 461.73 | 803.24 |
| 2002 | 3479.13 | 719.42 | 381.91 | 421.16 | 243.48 | 352.55 | 507.16 | 853.46 |
| 2003 | 4096.50 | 837.24 | 450.23 | 485.05 | 281.49 | 411.74 | 590.76 | 1039.99 |
| 2004 | 4563.96 | 938.00 | 495.90 | 528.05 | 312.21 | 451.57 | 649.31 | 1188.92 |
| 2005 | 5436.70 | 1146.47 | 566.70 | 599.41 | 376.44 | 519.93 | 755.62 | 1472.13 |
| 2006 | 6108.38 | 1288.76 | 633.53 | 666.01 | 428.81 | 583.65 | 852.48 | 1655.14 |
| 2007 | 6531.10 | 1392.01 | 676.34 | 696.40 | 463.74 | 609.99 | 906.38 | 1786.25 |
| 2008 | 6998.28 | 1517.54 | 718.40 | 728.45 | 515.83 | 644.30 | 969.04 | 1904.71 |
| 2009 | 7544.90 | 1624.49 | 775.05 | 798.30 | 562.53 | 707.13 | 1050.15 | 2027.24 |
| 2010 | 8255.42 | 1797.20 | 842.24 | 857.27 | 645.66 | 775.00 | 1153.96 | 2184.08 |
| 2011 | 9203.94 | 2034.41 | 956.37 | 880.08 | 836.84 | 855.11 | 1326.43 | 2314.70 |
| 2012 | 9392.04 | 2070.30 | 976.45 | 905.84 | 846.57 | 877.53 | 1354.86 | 2360.50 |
| 2013 | 9435.14 | 2040.80 | 992.97 | 942.87 | 932.23 | 884.33 | 1379.38 | 2262.56 |
| 2014 | 9627.14 | 2060.61 | 1019.12 | 958.79 | 955.57 | 905.33 | 1408.05 | 2319.68 |
| 2015 | 9094.56 | 1955.65 | 962.94 | 903.13 | 884.74 | 837.85 | 1316.03 | 2234.21 |
| 2016 | 9370.32 | 1955.91 | 995.40 | 934.62 | 907.05 | 863.46 | 1355.86 | 2318.02 |
| 2017 | 9531.09 | 1998.36 | 1022.76 | 904.80 | 999.15 | 909.80 | 1395.81 | 2300.41 |
| Accumulated value | 125,081 | 26,714 | 13,173 | 12,997 | 10,649 | 11,839 | 17,909 | 31,759 |
| Increment | 6336 | 1331 | 666 | 504 | 769 | 580 | 919 | 1566 |
| Chang rate (%) | 198.33 | 199.51 | 186.61 | 125.90 | 334.83 | 175.65 | 193.03 | 213.38 |

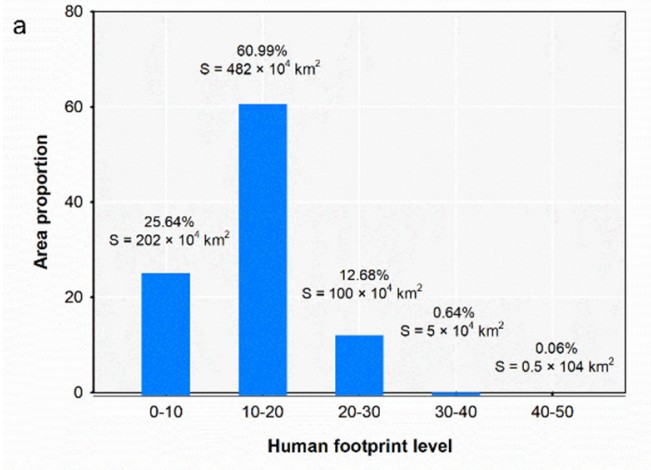
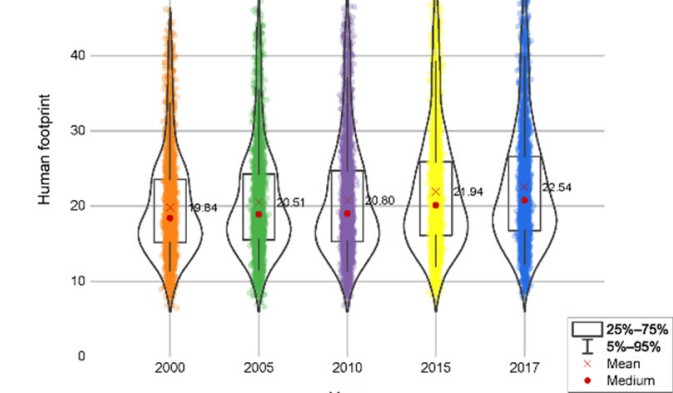

**Figure 1.** *Cont.*

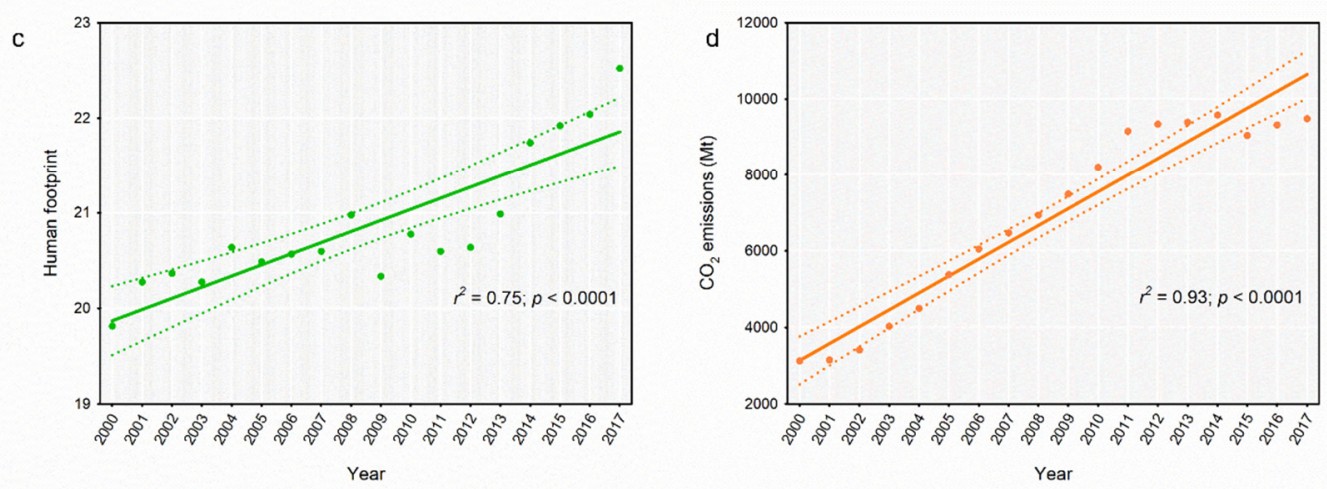

**Figure 1.** Features of human footprint and $CO_2$ emissions in China including (**a**) area of different human footprint classes, (**b**) variation in the average human footprint in different years, (**c**) regression analysis of human footprint and time, and (**d**) regression analysis of $CO_2$ emissions and time.

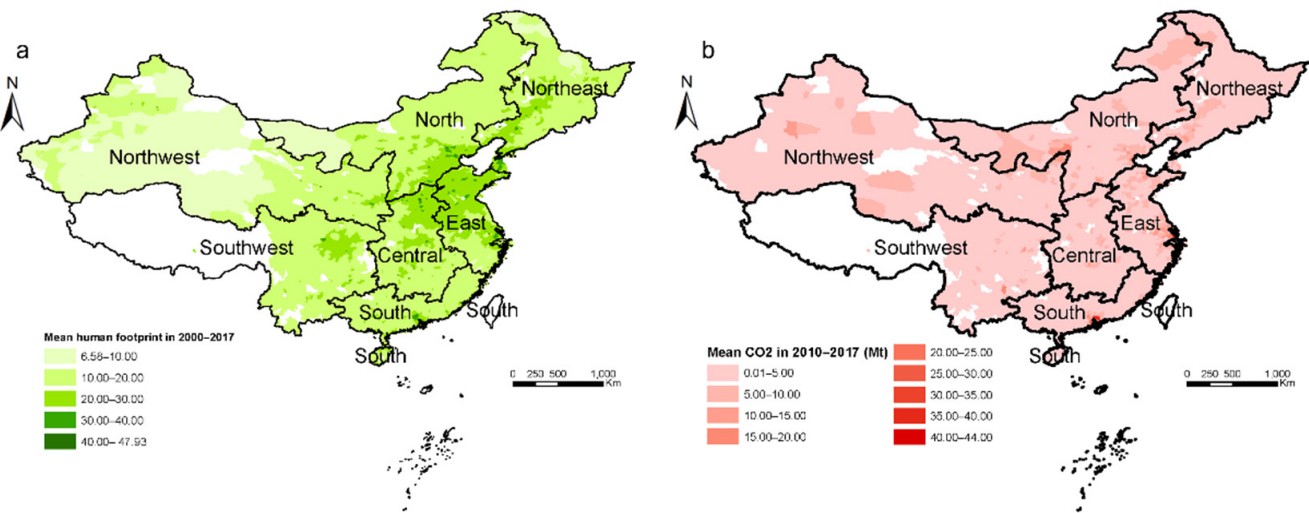

**Figure 2.** Spatial pattern of the average (**a**) human footprint and (**b**) $CO_2$ emissions in different regions in China during 2000–2017.

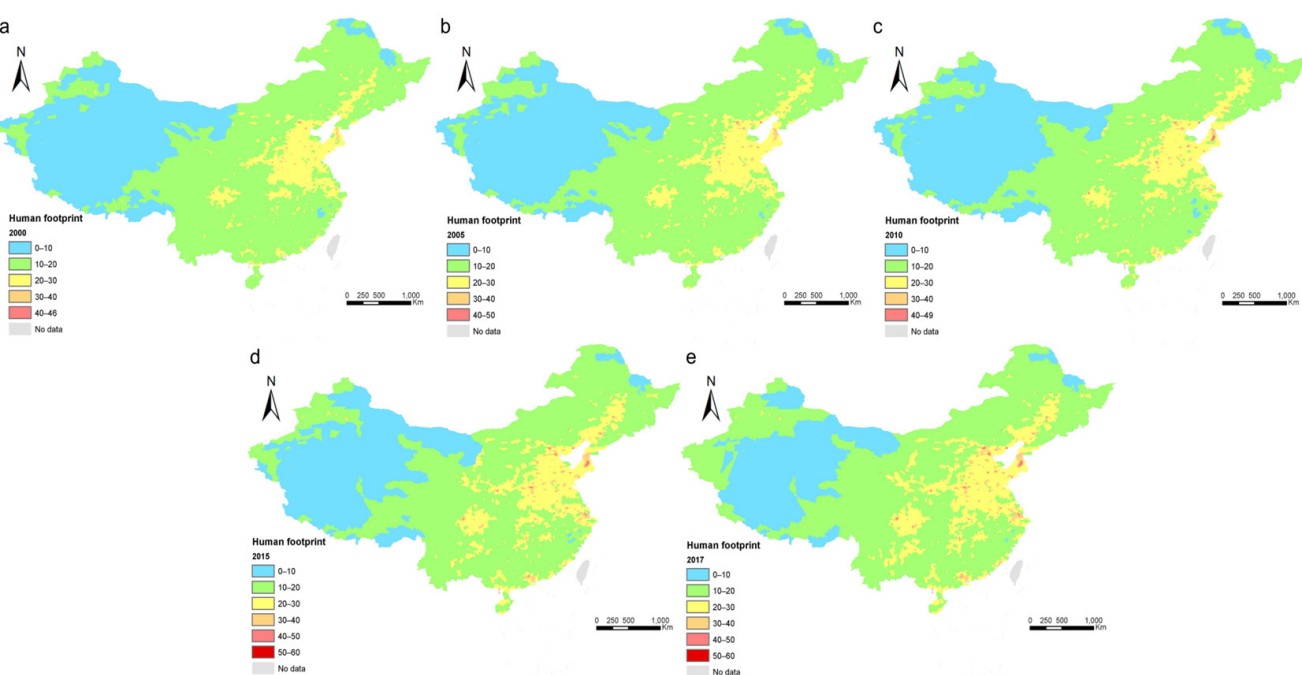

**Figure 3.** Spatio-temporal pattern of human footprint in China in (**a**) 2000, (**b**) 2005, (**c**) 2010, (**d**) 2015, and (**e**) 2017.

### 3.2. Spatio-Temporal Dynamics of $CO_2$ Emission Fluxes in China

China emitted a total of 125,081 Mt $CO_2$ during 2000–2017 (Table 2). $CO_2$ emissions in China gradually increased from 3194.81 in 2000 to 9531.09 Mt in 2017 (an increase of 198.33%). There was pronounced spatial variation in $CO_2$ emission fluxes in different regions (Figure 2b). $CO_2$ emissions contributed by eastern and northwestern China were the highest (31,759 Mt) and lowest (11,839 Mt), respectively (Table 2; Figure 4). $CO_2$ emission fluxes in China increased by 6336 Mt; the largest increase was observed in eastern China (1556 Mt), and the lowest increase was observed in northeastern China (504 Mt). $CO_2$ emissions in all regions of China increased by more than 100%. The rate of change in $CO_2$ emissions for northwestern (334.83%), eastern (213.38%), and northern China (199.51%) exceeded the national average (Table 2). The high (30–60), moderate (10–30), and low (0–10) $CO_2$ emissions in China increased 122.05, 1625.53, and 4588.70 Mt, respectively (Table S2; Figure S3). The high $CO_2$ emissions in southern (44.03 Mt) and eastern China (43.77 Mt) increased the most. Some of the low $CO_2$ emission areas were converted into high $CO_2$ emission areas (Table S2; Figure 5). $CO_2$ emissions were shown to have peaked in 2011 (+948.52 Mt) and were lowest in 2015 (−532.58 Mt) (Figure 6a). There was a gradual decrease in the magnitude of the increases of average $CO_2$ emissions in China during 2000–2017.

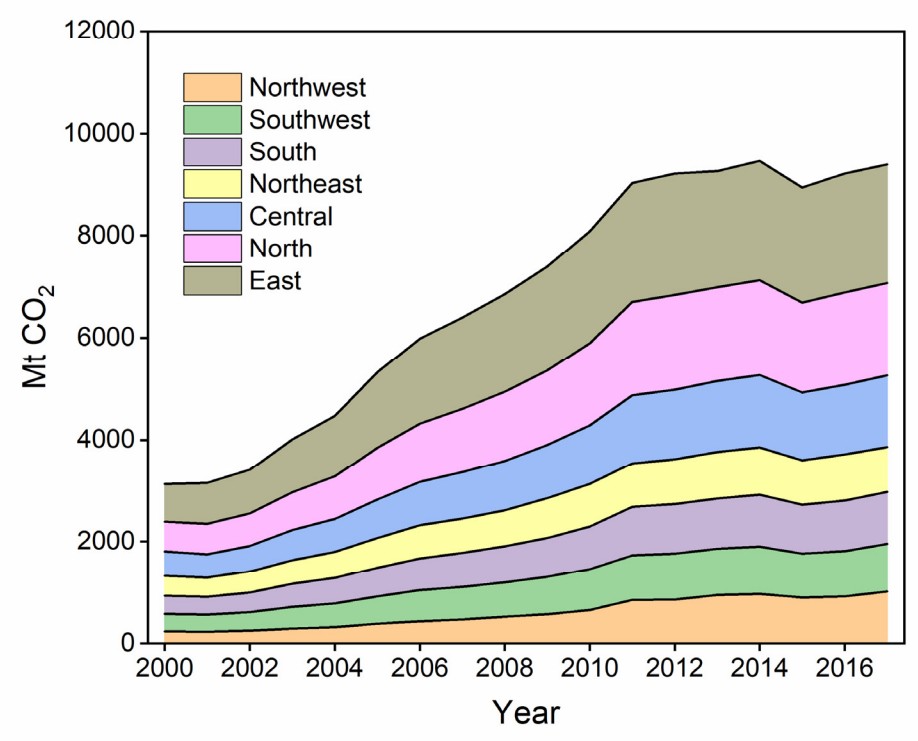

**Figure 4.** Stacked area map of annual $CO_2$ emissions in different regions in China during 2000–2017.

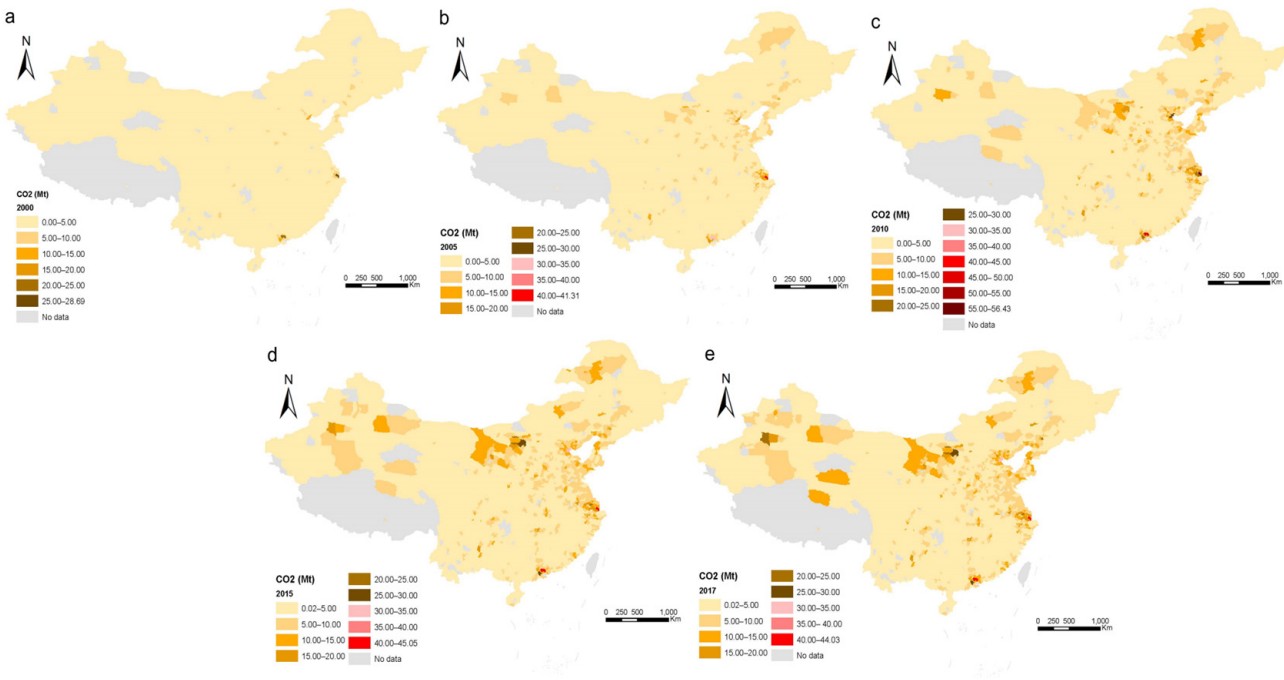

**Figure 5.** Spatio-temporal pattern of $CO_2$ emissions in China in (**a**) 2000, (**b**) 2005, (**c**) 2010, (**d**) 2015, and (**e**) 2017.

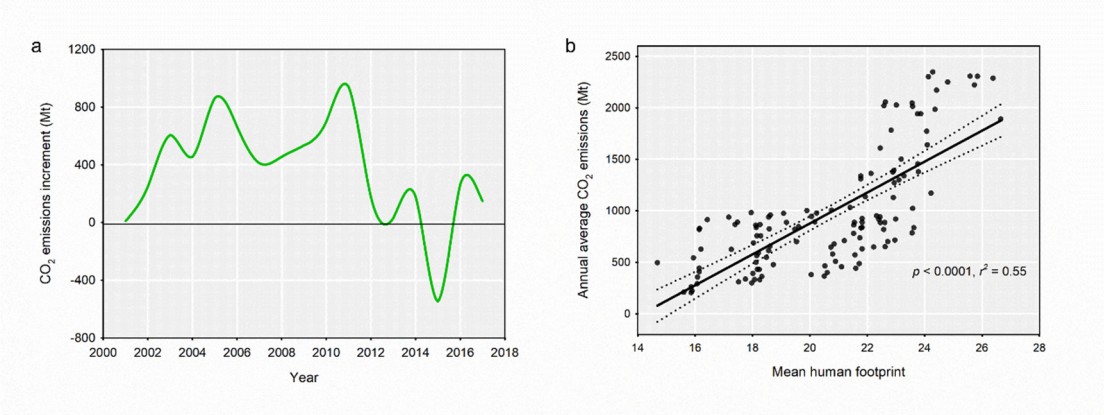

**Figure 6.** Trends in the (**a**) annual increase in $CO_2$ emissions and the (**b**) relationship between the average human footprint and average annual $CO_2$ emissions.

$CO_2$ emissions were higher in the provinces of Shandong (577.08 Mt y$^{-1}$), Jiangsu (497.00 Mt y$^{-1}$), and Hebei (492.35 Mt y$^{-1}$), which accounted for 23% of total emission fluxes in China (Figure S4). Global Moran's *I* was calculated to measure the spatial autocorrelation in $CO_2$ emissions. Its value was 0.14 ($p < 0.01$), which indicates a significant positive spatial autocorrelation in $CO_2$ emissions. The local Moran's *I* was used to further study the spatial characteristics of $CO_2$ emissions in 2000 (Figure 7a) and 2017 (Figure 7b) in China. There was a significant expansion of high-high cluster areas from 2000 to 2017, especially into eastern and northern China. The low–low cluster areas were mainly located in southwest China.

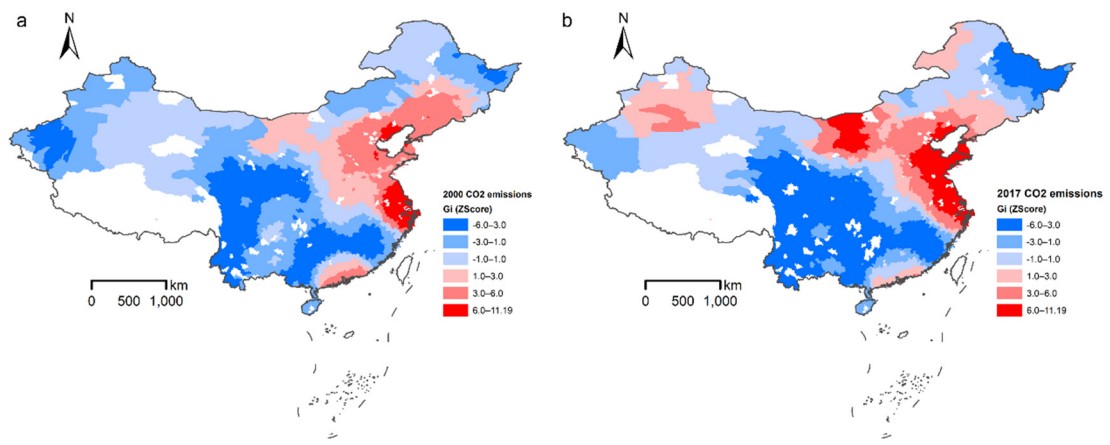

**Figure 7.** Spatial autocorrelation analysis (local Moran's *I* clustering) of $CO_2$ emissions in China in (**a**) 2000 and (**b**) 2017. Red and blue areas indicate positive and negative spatial correlation, respectively.

*3.3. Correlations between $CO_2$ Emissions and Human Pressures*

HF and $CO_2$ emissions in different regions of China were significantly positively correlated based on a Spearman's correlation coefficient for ranked data (correlation coefficient = 0.49, $p < 0.0001$) (Figure 6b). HF ($n = 18$; $r^2 = 0.75$; $p < 0.0001$) and $CO_2$ emissions ($n = 18$; $r^2 = 0.93$; $p < 0.0001$) in China increased significantly over time during the period 2000–2017 (Figure 1c; Figure 1d). The HF was highest (24.51) and lowest (16.29) in eastern and northwestern China, respectively (Table 2). $CO_2$ emissions were highest (1484 Mt) and lowest (592 Mt) in eastern and northwestern China, respectively (Table 2). During this 18-year period, the annual $CO_2$ emissions in each region were significantly positively correlated with HF ($n = 126$; $r^2 = 0.55$; $p < 0.0001$).

## 4. Discussion

### 4.1. Characteristics of Human Pressures in China

The human footprint provides a spatially explicit and temporally consistent quantitative measure of the magnitude of human pressures on the environment. Human pressures alter natural environments and harm natural systems [50]. These pressures include land-use cover (e.g., agricultural land and construction land), transportation (e.g., roads, railways, and navigable waterways), agricultural land and construction land, and trade (e.g., imports, exports, and international exports of energy-intensive industries) [51]. Human pressures (indicated by HF) could also be frequent, small in magnitude, and thus overlooked, such as the effects of extensive road systems and pasture lands. Spatial patterns of HF and human pressures were closely related. China has been the world's second largest economy since 2010. Spatial heterogeneity in regional human activities leads to spatial heterogeneity in human pressures in China. Eastern China is located between the mainland and the sea; it is characterized by gentle terrain and features agriculturally productive soils. In addition, this region is rich in aquatic products, oil, iron ore, salt, and other resources, the labor force is large, and the industrial and agricultural base is strong. By the end of 2017, eastern China accounted for only 5.25% of the country's area, but 23.28% of its population, 32.70% of its GDP, 32.99% of its road area, and 33.01% of its construction land. The average urbanization rate of eastern China was 64.56%, which was higher than the national rate of 58.52%; Shanghai had the highest urbanization rate in the country (87.70%). The high density of the population in this region has increased the demand for farmland, housing, and energy [52]. The extraction of environmental resources and the discharge of pollutants has had deleterious effects on the environment and has resulted in the loss of biodiversity and other ecosystem services, soil degradation, and the disruption of hydrological cycles [53]. Humans are constantly modifying the environment to accommodate their needs, yet this land-use change is also an important driver of human stress [54]. Over the past decades, China has experienced varying extents of urbanization. Intense urbanization improves social development in various ways but often creates substantial human pressures that could have negative effects on human welfare [55].

Human pressures in China have expanded over time. The main contributors to the HF model were construction land, areas of high population density, and road surroundings. From 2000–2017, China's population (1.27–1.40 billion, +0.6%/year), GDP (12.7–80.3 trillion, +31.0%/year), urbanization rate (36.0%–58.5%, +3.5%/year), farmland area (128.3–134.9 square kilometer, +0.3%/year), road area (2000–8000 square kilometer, +16.7%/year), livestock output (0.7–2.9 trillion, +16.5%/year) increased rapidly, which may be the reason that human pressure has increased by 13.6%. Human pressures from eastern, central, northeastern, and southwestern China have been increasing significantly over time, and the potential environmental implications caused by expanding human pressures associated with economic development require increased attention [56]. The control of these high-pressure areas is needed to relieve anthropogenically induced environmental damage.

### 4.2. $CO_2$ Emissions in China

Given China's large size and the heterogeneity in its economic development, lifestyles, resources, and economies vary among the provinces [57]. Eastern and northern China are the main contributors to $CO_2$ emissions and account for 45.39% of total emissions in China based on the 18-year average. The provinces of Shandong (2017 GDP: 1.12 trillion USD) and Jiangsu (2017 GDP: 1.38 trillion USD) in eastern China are two of the most developed regions in China and account for approximately 20% of the national total GDP. They also account for 15.71% of total $CO_2$ emissions given their large industrial areas and advanced technology. In contrast, the provinces of Inner Mongolia, Shanxi, and Hebei in northern China are less developed regions that account for only 7.86% of the country's GDP but 16.94% of its $CO_2$ emissions. Similar patterns were observed for northeastern and northwestern China.

$CO_2$ emissions continuously increased from 2000 to 2014 and declined in 2015–2017 by −5.52%, −3.10%, and −1.85% from the previous year, respectively. The economy of China has grown rapidly since its reform and opening in the 1970s. This economic expansion has led to the emission of a large amount of $CO_2$. From 2000 to 2017, China's GDP increased by 724.83% from 1.55 trillion US dollars (USD) to 12.80 trillion USD. Infrastructure construction and energy consumption have been the major drivers of the rapid growth of China's economy and emissions since 2002 [56]. The economy relies on carbon-intensive industries such as thermal power generation, steel, cement, and vehicle production. Due to increased greenhouse gas emissions, a tight energy supply, and severe air pollution, the government of China has begun to implement a series of strategies to conserve energy and mitigate emissions. The 11th and 12th Five-Year Plan (2006–2015) strategies have reduced emissions of $CO_2$ and coal by 3 billion tons and 1.4 billion tons, respectively [58]. From 2015–2017, the effect of slowing population growth (1.38–1.40 billion, +0.5%/year) and the optimization of industrial structure (2014, +4% $CO_2$) may result in a decrease in carbon emissions reduction and air pollution improvement. According to the report of the Netherlands Environmental Assessment Agency (PBL), China's carbon emissions increased by only 0.9% in 2014, despite its economic growth of 7% [59]. The slowdown in economic growth, coupled with the shift to cleaner energy and the reduction of energy intensive manufacturing has reduced the energy intensity of the country's economy. Control of the growth of $CO_2$ emissions in economically underdeveloped regions such as northwestern China will require the restructuring and upgrading of industry and industrial technology, respectively.

### 4.3. Impacts of Human Pressures on $CO_2$ Emissions

The $CO_2$ emissions per unit of economic output and per capita in the northern provinces are mostly higher than the national average, which indicates that economic development in northern China disproportionately contributes to $CO_2$ emissions compared with other regions of the country (Figure 8). For example, Shanghai led the country in $CO_2$ emissions per capita in 2000 (7.91 t per capita$^{-1}$), while Ningxia was first in 2017 (25.16 t per capita$^{-1}$) and has seen its $CO_2$ emissions increase more than three-fold since 2000 (5.97 t per capita$^{-1}$). The population and GDP are higher, and industrial development is more advanced in Shanghai compared with Ningxia. Several factors have caused the $CO_2$ emissions per capita to stabilize (2017, Shanghai, 7.95 t per capita$^{-1}$). The mismatch in the increase in GDP and technology levels in western China is an important factor affecting $CO_2$ emissions per capita. The high degree of economic development and the large human population in eastern China are the main reasons for the expansion of the high–high clustering area of $CO_2$ emissions. However, increasing energy consumption in northern China is the main factor underlying the emergence of the high–high clustering region [60]. Three northern provinces were among the top five emitters in 2000 (Figure 8a); in 2017 however, the top five emitters in the country were all northern provinces (Figure 8b). The carbon intensity ($CO_2$ per unit of GDP) of China decreased from 2000 to 2017 due to changes in economic development. The less economically developed provinces were mainly located in northern and western China. The carbon intensity of these regions mostly exceeded the national average in 2000 (49.86 t per USD$^{-1}$) (Figure 8c) and 2017 (17.00 t per USD$^{-1}$) (Figure 8d). Heavy industries are central to the economies of the northern provinces, and these products are mainly exported to other provinces [61]. Such provinces do not have adequate human resources to upgrade their technologies and equipment, and the economies of these provinces are mainly based on energy-intensive industries, such as mining, metals, electricity, and chemical products; consequently, the resource use efficiency of these provinces is low, and their greenhouse gas emissions are high [56]. China's economy is largely dependent on primary energy resources, and these resources are mainly located in less developed regions. Technology transfer and optimization of the industrial structure are important for reducing $CO_2$ emissions in less developed regions [62].

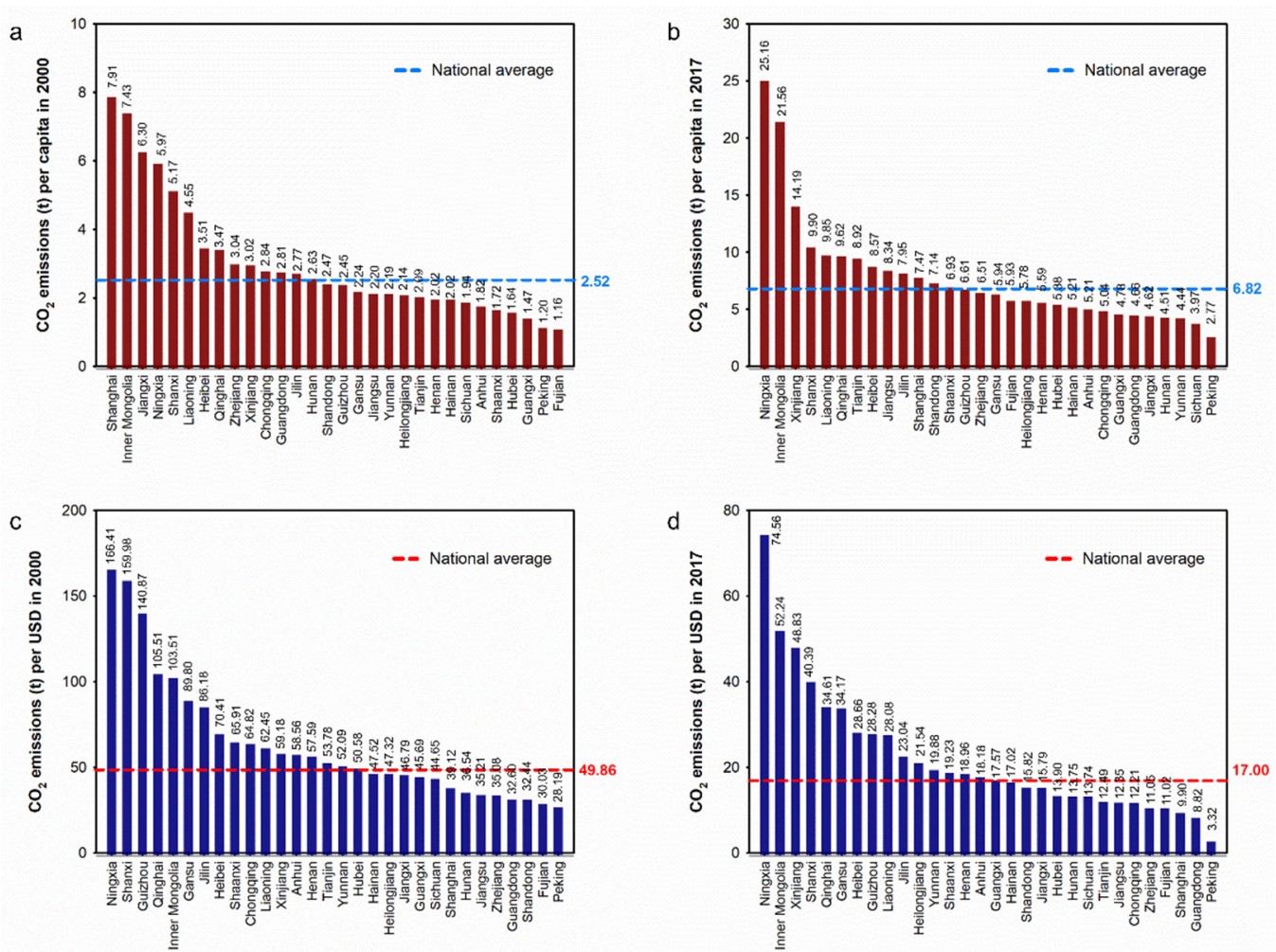

**Figure 8.** $CO_2$ emissions per capita (t) in (**a**) 2000 and (**b**) 2017 in each provincial administrative unit. $CO_2$ emissions per USD (g) in (**c**) 2000 and (**d**) 2017. The dotted line indicates the national average.

As increased human pressures lead to increased $CO_2$ emissions, spatial heterogeneity in the strength of human pressures leads to spatial heterogeneity in $CO_2$ emissions. Northern China is the main region contributing to increases in $CO_2$ emissions in response to human pressures. Thermal power generation is a major source of $CO_2$ emissions but an unsustainable form of energy generation in eastern and southern China. Northern China is rich in coal resources. The integrated development of coal and electricity has become widespread in China, and most thermal power generation units are concentrated in Inner Mongolia, Shanxi, and Xinjiang provinces [63], which has made northern China a major source of $CO_2$ emissions. Therefore, special effort is needed to control the intensity of human activities and greenhouse gas emissions in these regions. Modifications of the energy structure would be an effective approach to the reduction of $CO_2$ emissions in these regions [64]. Controlling the rate of increase in human pressures can also contribute to the mitigation of $CO_2$ emissions. There is thus a need to reduce the effect of human pressures on the environment, especially the deleterious effects of the human population, land-use structure, and urban construction.

One of the most important factors driving the reduction in energy consumption and $CO_2$ emissions in China is the increase in high-quality development. Photovoltaic (PV) power is considered one of the most promising low-carbon energy generation approaches in China [65]. By the end of 2015, the cumulative installed capacity of PV generation reached approximately 43 million kilowatts, which made China the largest solar power producer in the world [66]. By the end of 2030, the installed PV capacity will reach 100 million

kilowatts, which is equivalent to building more than 30 fewer large coal power plants. At the Climate Ambition Summit in 2020, China's national leader proposed that the total installed capacity of wind and solar power would reach more than 1.2 billion kilowatts by 2030. Ultra-high voltage (UHV) technology makes significant contributions to the reduction of $CO_2$ emissions [67]. China has been committed to developing UHV technology for many years. As of 2019, more than 20 UHV transmission lines have been built in China [68]. In addition, the Chinese government has been vigorously promoting the use of new energy vehicles (NEVs). Annual sales of NEVs were 1.367 million in 2020, which is more than 160 times the number of NEVs sold in 2011 (8159) [69]. Direct policy support has promoted the rapid development of China's NEV market and has made large contributions to energy saving and emission reductions.

## 5. Conclusions

Along with the development of economy and the acceleration of urbanization, human pressure on the environment is increasing. An understanding of the spatiotemporal dynamics in $CO_2$ emissions under changing human pressures is essential for designing sustainable environmental strategies. Our findings have implications for the development of mitigation policies for $CO_2$ emissions by local governments. The main findings and policy implications are manifold. First, carbon emissions in China are still on the rise; there is thus a need to strengthen the implementation of $CO_2$ reduction measures. Second, $CO_2$ emissions in China are unevenly distributed spatially (generally higher in the south and east and lower in the north and west), indicating that the government needs to optimize the regional allocation of energy. Third, increased human pressures have increased the amount of $CO_2$ emissions, and northern China is the main region driving this pattern. Given that little can be done to alter the current economic trend, the impacts of human activities on $CO_2$ emissions can be reduced by optimizing land use, population density, and grazing density. Fourth, $CO_2$ emissions associated with anthropogenic activities are decreasing. High-quality development measures and strong national macroeconomic control instruments are needed to achieve China's goal of carbon neutrality. We believe that China is on track to meet its carbon reduction commitments on time.

Remote sensing technology has expanded peoples' abilities to understand their living environment, and has the advantages of qualitative accuracy, large observation range, high spatial resolution, simple acquisition, and strong data consistency. However, government-based datasets have certain advantages in terms of quantitative analysis. Our next goal is to conduct a bottom-up carbon footprint coupled with remote sensing data and government-based datasets for an analysis of the impact of human activities on carbon emissions.

**Supplementary Materials:** The following supporting information can be downloaded at: https://www.mdpi.com/article/10.3390/rs15020426/s1. Figure S1. Analytical scope of CO2 emissions including (a) spatial distribution diagram of the monitoring area at the county level and (b) spatial distribution of regions in China used for estimating CO2 emissions. Figure S2. Annual CO2 emissions by seven regions in China. Figure S3. Annual CO2 emissions (g) per USD in China during 2000–2017. National average is 1.45 g CO2 per USD based the 18-year period. Figure S4. Average CO2 emissions in China in each provincial administrative unit. Table S1. Spatio-temporal characteristics of the human footprints in China. Table S2. Spatio-temporal characteristics of CO2 emissions.

**Author Contributions:** Y.L. and W.M. conceived the study and wrote the paper. L.J. and Y.Z. completed some of the statistical analyses. Q.H. and Y.W. provided suggestions for manuscript revision. Y.B. provided feedback on various drafts of the manuscript. All authors have read and agreed to the published version of the manuscript.

**Funding:** This work was supported by the National Natural Science Foundation of China (No. 31971477; No. 42177057), the Key Laboratory of Algal Biology, Institute of Hydrobiology, Chinese Academy of Sciences (202205), Taiyuan University of Science and Technology Introduction of Talent Start-up Fund (2020051), and Incentive Fund for Outstanding Doctors Working in Shanxi (20212071).

**Data Availability Statement:** All the original data can be obtained from given data sources. The full dataset, including the spatio-temporal maps of $CO_2$ emissions and human footprint index used in this study is available. All datasets generated during this study are available from the corresponding author on reasonable request.

**Conflicts of Interest:** The authors declare that they have no known competing financial interest or personal relationships that could have influenced the work reported in this paper.

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
