# Peer review of "Change of Human Footprint in China and Its Implications for Carbon Dioxide (CO2) Emissions"

_remotesensing, doi:10.3390/rs15020426_

Round 1
Reviewer 1 Report
The manuscript discusses the connection between human effects and footprint changes in China with change of CO2 emissions from 2000-2017. The investigation itself sounds interesting and meaningful, however the manuscript was not properly written in its prescribed format.
Major Changes needed
(1) Introduction: This manuscript discusses the impacts of rapid economic development towards CO2 emission, and the spatial variability of CO2 emission within different emission clusters in China. However, the authors should also compare their finding / observations with previous literature finding, e.g., major emission clusters and reasoning behind, future emission scenario etc., for example the followings:
https://www.sciencedirect.com/science/article/pii/S1462901118312905
https://www.mdpi.com/1996-1073/15/17/6210
https://www.belfercenter.org/sites/default/files/files/publication/Emissions%202018.pdf
(2) Introduction: About the choice of datasets, the authors have used all remote sensed datasets for land-use cover, roads, railways, human population density etc. Why don't simply adopt some datasets obtained from local monitoring network (if they are available)? Government-based datasets should be more fine-scaled, sometimes remote sensing datasets are of coarse spatial resolutions, and might not give reasonable results.
(3) Ordering of Manuscript: The Theories, Definitions and Methods should be put in Section 2, after Introduction, but before Results. Please formulate and re-order the entire manuscript again.
(4) Please provide the definition of HF in the Section of "Methods / Theories", and explain how it was being adopted in the past studies of human footprints towards CO2 emission etc.
(5) Section 2.1. Spatio-temporal pattern of human footprint in China - What are the historical reasons or factors that constitute to such temporal changes? The authors have only mentioned the trends, but provided no clues nor explanations for all these changes.
(6) Figure 1(c) and 1(d): What are the main implications / deductions that can be obtained from these statistical plots? The authors should give some explanation within different parts of the manuscript.
(7) In the newly added Section 2 - "Methods / Theories", the authors should explicitly lay down the statistical metrics adopted throughout the entire study, and explain why those statistical metrics were selected for validation / obtaining correlation.
(8) Section 2.2 - the authors should present all these results in a more systematic manner, perhaps put it in tabular form, or in spatial plots.
(9) Line 172: For obtaining r value and p value of HF, how many data points have the authors considered to reach such a good statistical performance?
(10) Section 3.1 - Characteristics of human pressures in China - The authors have not provided proper supporting evidence to validate their claim, some statistics are much preferred in this Section.
(11) In Section 3.1, the authors mentioned that different human pressures have altered natural environments and harmed natural systems. Can the authors explain fine details and the effects of each factor towards potential environmental changes? It's a bit vague for now.
(12) Section 3.2 (Lines 218-231): The authors are recommended to show these numerical figures / statistics on a map / table for easy comparison. It's hard to conduct comparison now.
(13) Lines 286-the end of Section 3: The authors have mentioned and discussed some conceptual ideas of reducing CO2 emission within the environment, however, proper plans or approaches, or even government-based methods and citizen-based methods should be explained in details in Discussion section. Currently, there are not much scientific insights, and the conclusion is rather loose, because it is not derived from statistical figures / metrics / comparison, but from qualitative perspectives only.
(14) A proper Conclusion section is missing in the current manuscript.
Grammatical Error / Typo
Line 26: slowed down
Line 92: slowed down
Line 127 (Caption of Table 1) Average human footprint index in
Line 170: Do you mean "country" rather than "county"?
Lines 234-235: Due to increased greenhouse gas emissions,...
Line 241: require the restructuring and upgrading of...
Lines 268-269: largely depending on
Line 270: structure are important
Author Response
Reviewer #1:
Special thanks to you for your good comments. Detailed comments and our detailed responses/corrections are listed as below:
- Introduction: This manuscript discusses the impacts of rapid economic development towards CO2 emission, and the spatial variability of CO2 emission within different emission clusters in China. However, the authors should also compare their finding / observations with previous literature finding, e.g., major emission clusters and reasoning behind, future emission scenario etc., for example the followings:
Response: Thanks for your comment. We have added more details for future emission scenario and macro policy benefits on Line 73-77, Page 2 in the revised manuscript.
“Besides, key variables including technology, infrastructure, environment and finance have important practical significance to achieve carbon neutrality. For example, Labzovskii et al. (2017) projected that beneficial policy would result in 24%, 80%, 166% less CO2 emissions in East Asia by 2020, 2025 and 2030, respectively.”
References:
Mi, Z. et al. Socioeconomic impact assessment of China’s CO2 emissions peak prior to 2030. J. Clean. Prod. 142, 2227–2236 (2017).
Pflugmann, F., Blasio, N. D. The geopolitics of renewable hydrogen in low-carbon energy markets. Geopolitics, History and International Relations 12, 9–44 (2020).
Li et al. The carbon emission reduction effect of city cluster—evidence from the Yangtze River Economic Belt in China. Energies 15, 6210 (2022).
Labzovskii, L. D. et al. What can we learn about effectiveness of carbon reduction policies from interannual variability of fossil fuel CO2 emissions in East Asia? Environ. Sci. Policy 96, 132–140 (2019).
- Introduction: About the choice of datasets, the authors have used all remote sensed datasets for land-use cover, roads, railways, human population density etc. Why don't simply adopt some datasets obtained from local monitoring network (if they are available)? Government-based datasets should be more fine-scaled, sometimes remote sensing datasets are of coarse spatial resolutions, and might not give reasonable results.
Response: Thanks for your comment and suggestion. In this study, an understanding of the spatiotemporal dynamics in CO2 emissions under changing human pressures is essential for designing sustainable environmental strategies. Government-based datasets have certain advantages in qualitative analysis. Now we are making a bottom-up carbon footprint based on government-based datasets. Then we will continue to analyze the impact of China’s high-quality development on CO2 emissions.
- Ordering of Manuscript: The Theories, Definitions and Methods should be put in Section 2, after Introduction, but before Results. Please formulate and re-order the entire manuscript again.
Response: Thanks for your comment. We have adjusted the order of each part in the revised manuscript.
- Please provide the definition of HF in the Section of "Methods / Theories", and explain how it was being adopted in the past studies of human footprints towards CO2 emission etc.
Response: Thanks for your comment. We have provided the definition of HF in the Section of “Methods” in the revised manuscript.
“The human footprint (HF) is a global map of human in fluence on the land surface.”
References:
Sanderson, E. W. et al. The human footprint and the last of the wild. Bioscience 52, 172–173 (2002).
- Section 2.1. Spatio-temporal pattern of human footprint in China - What are the historical reasons or factors that constitute to such temporal changes? The authors have only mentioned the trends, but provided no clues nor explanations for all these changes.
Response: Thanks for your comment. Human pressures in China have expanded over time. Human pressures from East, Central, Northeast, and Southwest China have been increasing significantly over time. The main contributors to the HF model were construction land, areas of high population density, and road surroundings. Spatio-temporal pattern of HF was closely related to human activities. Spatial heterogeneity in regional human activities leads to spatial heterogeneity in human pressures in China. For example, the urbanization rate, industrialization level, population density, and land-use change of East China are higher/stronger than that of other place, where the environment pressures are highest. Humans are constantly modifying the environment to accommodate their needs, yet this land-use change is also an important driver of human stress. Over the past decades, China has experienced varying extents of urbanization. Intense urbanization improves social development in various ways but often creates substantial human pressures that could have negative effects on human welfare. We have provided the explanations for spatio-temporal pattern of HF change in Section 4.1 in the revised manuscript.
“Human pressures (indicated by HF) could also be frequent, small in magnitude, and thus overlooked, such as the effects of extensive road systems and pasture lands. Spatial pat-terns of HF and human pressures were closely related. China has been the world’s second largest economy since 2010. Spatial heterogeneity in regional human activities leads to spatial heterogeneity in human pressures in China. East China is located between the mainland and the sea; it was characterized by gentle terrain and features agriculturally productive soils. In addition, this region is rich in aquatic products, oil, iron ore, salt, and other resources, the labor force is large, and the industrial and agricultural base is strong. Over the past decades, China has experienced varying extents of urbanization. Intense urbanization improves social development in various ways but often creates substantial human pressures that could have negative effects on human welfare.
Human pressures in China have expanded over time. The main contributors to the HF model were construction land, areas of high population density, and road surroundings. Human pressures from East, Central, Northeast, and Southwest China have been increasing significantly over time, and the potential environmental implications caused by expanding human pressures associated with economic development require increased attention.”
- Figure 1(c) and 1(d): What are the main implications / deductions that can be obtained from these statistical plots? The authors should give some explanation within different parts of the manuscript.
Response: Thanks for your comment. We have added some explanation about HF increase and CO2 emissions decrease on Line 292-301, Page 6-7; Line 324-331, Page 7 in the revised manuscript.
“Human pressures in China have expanded over time. The main contributors to the HF model were construction land, areas of high population density, and road surroundings. From 2000–2017, China’s population (12.7–14.0 billion, +0.6%/year), GDP (12.7–80.3 trillion, +31.0%/year), urbanization rate (36.0%–58.5%, +3.5%/year), farmland area (128.3–134.9 square kilometer, +0.3%/year), road area (2000–8000 square kilometer, +16.7%/year), livestock output (0.7–2.9 trillion, +16.5%/year) were increasing rapidly, which may be the reason why human pressure has increased by 13.6%.”
“The 11th and 12th Five-Year Plan (2006–2015) strategies have reduced emissions of CO2 and coal by 3 billion tons and 1.4 billion tons, respectively. From 2015–2017, effect of slowing down population growth (13.8–14.0 billion, +0.5%/year) and optimizing industrial structure (2014, +4% CO2) may result in a decrease in carbon emissions reducing and air pollution improvement. According to the report of Netherlands Environmental Assessment Agency (PBL), China’s carbon emissions increased by only 0.9% in 2014, despite its economic growth of 7%. The slowdown in economic growth, coupled with the shift to cleaner energy and the reduction of energy intensive manufacturing, has reduced the energy intensity of the country's economy.”
- In the newly added Section 2 - "Methods / Theories", the authors should explicitly lay down the statistical metrics adopted throughout the entire study, and explain why those statistical metrics were selected for validation / obtaining correlation.
Response: Thanks for your comment. We have explicitly laid down the statistical metrics adopted throughout the entire study in Section 2.8 in the revised manuscript.
2.8. Statistical analyses
“We performed all simple linear regression analysis, including the determination of 95% confidence intervals, with SigmaPlot 14.0 (Systat Software, USA). r2 indicates the fit of a one-dimensional linear regression, and p<0.0001 was the threshold for statistical significance. Standard deviations of CO2 emissions and HF by different regions were calculated using SigmaPlot 14.0 software. We evaluated bivariate comparisons of continuous data measurements using analysis of variance (ANOVA) tests. CO2 emissions and HF by different regions were expressed as average and standard error (SE). We tested the errors of HF calculation using a Monte Carlo simulation with 1,000 iterations. We used to make histograms, conduct linear regressions, and build line plots. We used Origin Pro 2021 (Microcal Software, USA) to make violin plots and stacked area plots. All the spatial pat-tern of CO2 emissions and HF in China was analyzed by ArcGIS 10.2 (ESRI, USA).”
- Section 2.2 - the authors should present all these results in a more systematic manner, perhaps put it in tabular form, or in spatial plots.
Response: Thanks for your comment. We have presented a stacked histogram of CO2 emissions in various regions of China (Fig. S2) in Supplementary Information.
Fig. S2 Annual CO2 emissions by seven regions in China.
- Line 172: For obtaining r value and p value of HF, how many data points have the authors considered to reach such a good statistical performance?
Response: Thanks for your comment. In order to analyze the change trend of HF and CO2 emissions over time, we have made a simple linear regression analysis of time with a total of 18 points of annual average. In the regression analysis of average human footprint and annual average carbon emissions, we used 126 points, including the average values of 7 regions in 18 years. The corresponding n value is marked on Line 256-262, Page 6 in the manuscript.
“HF (n = 18; r2 = 0.75; p < 0.0001) and CO2 emissions (n = 18; r2 = 0.93; p < 0.0001) in China increased significantly over time during 2000–2017.”
“During this 18-year period, the annual CO2 emissions in each region were significantly positively correlated with HF (n = 126; r2 = 0.55; p < 0.0001).”
- Section 3.1 - Characteristics of human pressures in China - The authors have not provided proper supporting evidence to validate their claim, some statistics are much preferred in this Section. In Section 3.1, the authors mentioned that different human pressures have altered natural environments and harmed natural systems. Can the authors explain fine details and the effects of each factor towards potential environmental changes? It's a bit vague for now.
Response: Thanks for your comment. We have provided some statistical data to explain our analysis results on Line 270-278, Page 6; Line 288-301, Page 6-7 in the revised manuscript.
“Human pressures (indicated by HF) could also be frequent, small in magnitude, and thus overlooked, such as the effects of extensive road systems and pasture lands. Spatial patterns of HF and human pressures were closely related. China has been the world’s second largest economy since 2010. Spatial heterogeneity in regional human activities leads to spatial heterogeneity in human pressures in China. East China is located between the mainland and the sea; it was characterized by gentle terrain and features agriculturally productive soils. In addition, this region is rich in aquatic products, oil, iron ore, salt, and other resources, the labor force is large, and the industrial and agricultural base is strong.”
“Over the past decades, China has experienced varying extents of urbanization. Intense urbanization improves social development in various ways but often creates substantial human pressures that could have negative effects on human welfare.”
“Human pressures in China have expanded over time. The main contributors to the HF model were construction land, areas of high population density, and road surroundings. From 2000–2017, China’s population (12.7–14.0 billion, +0.6%/year), GDP (12.7–80.3 trillion, +31.0%/year), urbanization rate (36.0%–58.5%, +3.5%/year), farmland area (128.3–134.9 square kilometer, +0.3%/year), road area (2000–8000 square kilometer, +16.7%/year), livestock output (0.7–2.9 trillion, +16.5%/year) were increasing rapidly, which may be the reason why human pressure has increased by 13.6%. Human pressures from East, Central, Northeast, and Southwest China have been increasing significantly over time, and the potential environmental implications caused by expanding human pressures associated with economic development require increased attention.”
- Section 3.2 (Lines 218-231): The authors are recommended to show these numerical figures / statistics on a map / table for easy comparison. It's hard to conduct comparison now.
Response: Thanks for your comment. We have added the corresponding statistics on Fig. 8 in the revised manuscript.
Fig. 8 CO2 emissions per capita (t) in (a) 2000 and (b) 2017 in each provincial administrative unit. CO2 emissions per USD (g) in (c) 2000 and (d) 2017. The dotted line indicates the national average.
- Lines 286-the end of Section 3: The authors have mentioned and discussed some conceptual ideas of reducing CO2 emission within the environment, however, proper plans or approaches, or even government-based methods and citizen-based methods should be explained in details in Discussion section. Currently, there are not much scientific insights, and the conclusion is rather loose, because it is not derived from statistical figures/metrics/comparison, but from qualitative perspectives only.
Response: Thanks for your comment. This study focused on the correlation between human pressure changes and carbon emission trends. Compared with the bottom-up statistical data, the disadvantage of remote sensing data is that it is unable to interpret the results from a more quantitative perspective. In the discussion section, we have added corresponding government statistics as a description on Line 270-278, Page 6; Line 288-301, Page 6-7; Line 324-331, Page 7. We are also working on a bottom-up model for calculating China’s carbon footprint and a landscape index model based on a 1 km grid to find more specific and direct reasons for China’s carbon emission reduction benefits.
“Human pressures (indicated by HF) could also be frequent, small in magnitude, and thus overlooked, such as the effects of extensive road systems and pasture lands. Spatial patterns of HF and human pressures were closely related. China has been the world’s second largest economy since 2010. Spatial heterogeneity in regional human activities leads to spatial heterogeneity in human pressures in China. East China is located between the mainland and the sea; it was characterized by gentle terrain and features agriculturally productive soils. In addition, this region is rich in aquatic products, oil, iron ore, salt, and other resources, the labor force is large, and the industrial and agricultural base is strong.”
“Over the past decades, China has experienced varying extents of urbanization. Intense urbanization improves social development in various ways but often creates substantial human pressures that could have negative effects on human welfare.”
“Human pressures in China have expanded over time. The main contributors to the HF model were construction land, areas of high population density, and road surroundings. From 2000–2017, China’s population (12.7–14.0 billion, +0.6%/year), GDP (12.7–80.3 trillion, +31.0%/year), urbanization rate (36.0%–58.5%, +3.5%/year), farmland area (128.3–134.9 square kilometer, +0.3%/year), road area (2000–8000 square kilometer, +16.7%/year), livestock output (0.7–2.9 trillion, +16.5%/year) were increasing rapidly, which may be the reason why human pressure has increased by 13.6%. Human pressures from East, Central, Northeast, and Southwest China have been increasing significantly over time, and the potential environmental implications caused by expanding human pressures associated with economic development require increased attention.”
“The 11th and 12th Five-Year Plan (2006–2015) strategies have reduced emissions of CO2 and coal by 3 billion tons and 1.4 billion tons, respectively. From 2015–2017, effect of slowing down population growth (13.8–14.0 billion, +0.5%/year) and optimizing industrial structure (2014, +4% CO2) may result in a decrease in carbon emissions reducing and air pollution improvement. According to the report of Netherlands Environmental Assessment Agency (PBL), China's carbon emissions increased by only 0.9% in 2014, despite its economic growth of 7%. The slowdown in economic growth, coupled with the shift to cleaner energy and the reduction of energy intensive manufacturing, has reduced the energy intensity of the country's economy.”
- A proper Conclusion section is missing in the current manuscript.
Response: Thanks for your comment. We have added the Conclusion in the revised manuscript.
“Along with the development of economy and the acceleration of urbanization, human pressure on the environment is increasing. An understanding of the spatiotemporal dynamics in CO2 emissions under changing human pressures is essential for designing sustainable environmental strategies. Our findings have implications for the development of CO2 emissions mitigation policies by the local government. The main findings and pol-icy implications are manifold. First, carbon emissions in China are still on the rise; there is thus a need to strengthen the implementation of CO2 reduction measures. Second, CO2 emissions in China are unevenly distributed spatially (generally higher in the south and east and lower in the north and west), indicating that the government needs to optimize the regional allocation of energy. Third, increased human pressures have increased the amount of CO2 emissions, and northern China is the main region driving this pattern. Given that little can be done to alter the current economic trend, the impacts of human activities on CO2 emissions can be reduced by optimizing land use, population density, and grazing density. Fourth, CO2 emissions associated with anthropogenic activities are de-creasing. High-quality development measures and strong national macroeconomic control instruments are needed to achieve China’s goal of carbon neutrality. We believe that China is on track to meet its carbon reduction commitments on time.”
- Line 26: slowed down
Response: Thanks for your comment. We have corrected this error on Line 29, Page 1 in the revised manuscript.
- Line 92: slowed down
Response: Thanks for your comment. We have corrected this error on Line 98, Page 3 in the revised manuscript.
- Line 127 (Caption of Table 1) Average human footprint index in
Response: Thanks for your comment. We have corrected this error in Table 1 on Line 535, 538 , Page 14-15 in the revised manuscript.
- Line 170: Do you mean "country" rather than "county"?
Response: Thanks for your comment. We have corrected this error “average CO2 emissions in China” on Line 243, Page 5 in the revised manuscript.
- Lines 234-235: Due to increased greenhouse gas emissions,...
Response: Thanks for your comment. We have corrected this error on Line 321-322, Page 7 in the revised manuscript.
- Line 241: require the restructuring and upgrading of...
Response: Thanks for your comment. We have corrected this error on Line 333, Page 7 in the revised manuscript.
- Lines 268-269: largely depending on
Response: Thanks for your comment. We have corrected this error on Line 361-362, Page 8 in the revised manuscript.
- Line 270: structure are important
Response: Thanks for your comment. We have corrected this error on Line 363, Page 8 in the revised manuscript.

Reviewer 2 Report
This manuscript used the human foot-print index (HF) to explore the impacts of human pressures on CO2 emissions in China from 2000 to 2017. There are interesting aspects of this study and such work is important, especially after China's carbon-neutral goal. Here, I have some comments for further improvement.
1. Line 344-345: Why the pressure score is assigned this way, and on what basis? How do you think about forest land? Sometimes they are huge carbon sinks that can reduce carbon emissions.
2. Line 407-415: What emission sources are included in the CO2 fluxes data, and whether land use change and disturbance (such as forest fire) are included?
3. Line 137-140: How do you divide up the areas? I suggest to list the provinces included in each region.
4. Figure 2 and 3: I suggest adding a North Arrow.
5. Line 146: The emission data is average or total value during 200-2017.
6. Figure 1: I found that HF decreased from 2010 to 2013, while CO2 emission actually increased. Why? If it is because of the high quality development in China, does it mean that the correlation between HF and CO2 emissions will fail in the future?
7. I think GDP data should have a good correlation with CO2 emissions and also reflect the human footprint to some extent. You also used a lot of GDP data to explain carbon emission trends in your discussion, but why is it not taken into account when calculating HF?
8. I cann't see the conclusion, does not need that?
Author Response
Reviewer #2:
Special thanks to you for your good comments. Detailed comments and our detailed responses/corrections are listed as below:
- Line 344-345: Why the pressure score is assigned this way, and on what basis? How do you think about forest land? Sometimes they are huge carbon sinks that can reduce carbon emissions.
Response: Thanks for your comment. We are sorry we didn’t explain it clearly. Corresponding explanations and references have been added to the manuscript. These pressures were weighted according to estimates of their relative levels of human pressure following Sanderson et al. 2002 and summed together to create the standardized human footprint for all non-Antarctic land areas. Land-use cover change is an important factor reflecting the degree of human activities. According to the interference and impact of different land-use types on environment, exchange between natural environmental factors (air, water, and soil), maintenance of natural balance, the construction land with impervious surface completely changed the natural habitat (assigned 10 score). The change of agricultural land to nature is less than that of construction land, and the impact on the environment mainly comes from fertilization and irrigation. Although intensive agriculture often results in whole-scale ecosystem conversion, we gave it a lower score than built environments because of less impervious cover (assigned 7 score). The impact of grassland on the environment is mainly represented by grazing (assigned 4 score). The forest is a natural and primitive ecosystem and land-use cover of natural surface is not changed (assigned 0 score). The assigned pressure score is based on Duan & Luo, 2020 and Venter et al, 2016. Sometimes they are indeed huge carbon sinks that can reduce carbon emissions. In this study, we only built a comprehensive indicator of human pressure on the environment before coupling carbon emission.
|
Land-use cover |
Characteristic |
score |
|
Grassland |
Land-use cover of natural surface is not changed but used |
4 |
|
Agricultural land |
Land-use cover of natural surface is changed by planting crops |
7 |
|
Construction land |
Artificial barriers-surface; water, nutrient, air, and heat retardation |
10 |
|
Forest |
Land-use cover of natural surface is not changed |
0 |
References:
Duan, Q. T., Luo, L. A dataset of human footprint over the Qinghai-Tibet Plateau during 1990–2017. (2021). National Tibetan Plateau/Third Pole Environment Data Center. DOI:10.11922/sciencedb.933. (in Chinese)
Sanderson, E. W. et al. The human footprint and the last of the wild. Bioscience 52, 172–173 (2002).
Venter, O. et al. Sixteen years of change in the global terrestrial human footprint and implications for biodiversity conservation. Nat. Commun. 7, 12558 (2016).
- Line 407-415: What emission sources are included in the CO2 fluxes data, and whether land use change and disturbance (such as forest fire) are included?
Response: Thanks for your comment. The CO2 fluxes data were calculated using provincial energy balance tables, where 17 types of fossil fuel used are considered, including raw coal, cleaned coal, other washed coal, briquettes, gangue, coke, coke oven gas, blast furnace gas, converter gas, other gases, other coking products, crude oil, gasoline, kerosene, diesel oil, fuel oil, naphtha, lubricants, paraffin, white spirit, bitumen asphalt, petroleum coke, other petroleum products, liquefied petroleum gas (LPG), refinery gas, and natural gas. In addition, carbon sequestration values of terrestrial vegetation were estimated. With regard to the concept of plant carbon sequestration capacity, it is a natural carbon sequestration process, which directly counteracts the processes of emitting CO2 into the atmosphere. The CO2 fluxes data do not directly assess the CO2 emissions caused by land use change and disturbance (such as forest fire). However, the impact of forest change on CO2 emissions is explained to some extent from the perspective of carbon sequestration of vegetation (Chen et al., 2020).
References:
Chen, J., et al. County-level CO2 emissions and sequestration in China during 1997–2017. Sci. Data, 7, 391 (2020). https://doi.org/10.1038/s41597-020-00736-3
- Line 137-140: How do you divide up the areas? I suggest to list the provinces included in each region.
Response: Thanks for your comment. The basic of the seven major physical geographical divisions in China is determined on the basis of science, integrating multiple dimensions such as history and nationality, and following relevant division principles. We have listed the provinces included in each region in Supplementary Information.
“Northeast provinces: Heilongjiang, Jilin, Liaoning
North provinces: Beijing, Tianjin, Hebei, Shanxi, Inner Mongolia
Northwest provinces: Gansu, Ningxia, Shan’xi, Qinghai, Xinjiang
East provinces: Shanghai, Jiangsu, Zhejiang, Anhui, Shandong
Central provinces: Hunan, Hubei, Henan, Jiangxi
Southwest provinces: Sichuan, Chongqing, Tibet, Yunnan, Guizhou
South provinces: Guangdong, Guangxi, Fujian, Hainan”
- Figure 2 and 3: I suggest adding a North Arrow.
Response: Thanks for your comment. We have added North Arrow in Figure 2, 3 and 5.
- Line 146: The emission data is average or total value during 2000-2017.
Response: Thanks for your comment. We are sorry that we don’t describe it clearly. We have modified it on Line 243, Page 5 in the revised manuscript.
“There was a gradual decrease in the magnitude of the increases of average CO2 emissions in China during 2000–2017.”
- Figure 1: I found that HF decreased from 2010 to 2013, while CO2 emission actually increased. Why? If it is because of the high-quality development in China, does it mean that the correlation between HF and CO2 emissions will fail in the future?
Response: Thanks for your comment. In a short period of time, HF and CO2 emission are not necessarily positive correlation. For example, HF decreased from 2010 to 2013, while CO2 emission actually increased. HF and CO2 emission have a significant positive correlation in a long term. In this study, we would like to express the view that although the human pressure continues to increase, the growth rate of CO2 emissions has slowed down significantly under China’s high-quality development.
- I think GDP data should have a good correlation with CO2 emissions and also reflect the human footprint to some extent. You also used a lot of GDP data to explain carbon emission trends in your discussion, but why is it not taken into account when calculating HF?
Response: Thanks for your comment. I quite agree with you. GDP data have a good correlation with CO2 emissions/human footprint to some extent. In this study, we mainly considered the night light index as an important basis for economic development. Moreover, GDP has not been found as the condition for evaluating human footprint, and the basis for its assignment is still unclear. When explaining the trend of carbon emissions, the quantitative GDP can well describe the data. Now, we are planning to analyze a bottom-up assessment of China’s carbon footprint using some datasets obtained from local monitoring network, like GDP, energy intensity, and agricultural emissions data, etc.
- I cann't see the conclusion, does not need that?
Response: Thanks for your comment. We have added the Conclusion in the revised manuscript.
“Along with the development of economy and the acceleration of urbanization, human pressure on the environment is increasing. An understanding of the spatiotemporal dynamics in CO2 emissions under changing human pressures is essential for designing sustainable environmental strategies. Our findings have implications for the development of CO2 emissions mitigation policies by the local government. The main findings and pol-icy implications are manifold. First, carbon emissions in China are still on the rise; there is thus a need to strengthen the implementation of CO2 reduction measures. Second, CO2 emissions in China are unevenly distributed spatially (generally higher in the south and east and lower in the north and west), indicating that the government needs to optimize the regional allocation of energy. Third, increased human pressures have increased the amount of CO2 emissions, and northern China is the main region driving this pattern. Given that little can be done to alter the current economic trend, the impacts of human activities on CO2 emissions can be reduced by optimizing land use, population density, and grazing density. Fourth, CO2 emissions associated with anthropogenic activities are de-creasing. High-quality development measures and strong national macroeconomic control instruments are needed to achieve China’s goal of carbon neutrality. We believe that China is on track to meet its carbon reduction commitments on time.”

Round 2
Reviewer 1 Report
Thanks the authors for revising this manuscript, it looks much clearer in terms of structure and technical details now. There are 6 more points to be further addressed before the publication of the current manuscript.
(1) Referring to point (2): About the appropriateness and technical deficiencies of the use of remotely sensed datasets, the authors should at least mention this in a humble manner, either at the beginning of Sec. 2, or at the end of this manuscript. They should include the response contents into the manuscript somehow - i.e., mentioning that government-based datasets have certain advantages in terms of quantitative analysis, and the next goal is to prepare a bottom-up carbon footprint and conduct corresponding analyses - some future goals have to be explicitly mentioned.
(2) Referring to point (2): Section 2.2 - Section 2.7 are all related to datasets adopted in this study. Some references should be added to explain how these types / categories of datasets were being adopted in the past for similar analyses / spatial analyses. This could enhance the confidence of usage of remotely sensed datasets. For example,
Section 2.2 (Land-use cover):
https://www.mdpi.com/2072-4292/13/16/3337
https://link.springer.com/article/10.1007/s12518-022-00441-3
Section 2.3 (Roads and railways):
https://www.mdpi.com/2220-9964/10/4/265
Section 2.4 (Population Density):
https://www.nature.com/articles/s41597-022-01675-x
Section 2.6 (Night-time lights):
https://iopscience.iop.org/article/10.1088/2515-7620/ab3d91/pdf
https://www.tandfonline.com/doi/full/10.1080/17538947.2021.1946605
Section 2.7 (CO2 fluxes):
https://www.sciencedirect.com/science/article/pii/S1001074221001431
(3) Referring to point (4): HF has now been explained, which sounds great, but some details of past studies between HF and CO2 emission figures should be provided as well.
(4) In "Section 1: Introduction", please include the scientific advancement and breakthrough made in current manuscript in an explicit manner. Further, please include a short paragraph describing the main role / idea of each Section.
(5) For "Section 2: Methods", I think it should be "2. Methods and Data", or the authors should separate Methods from Datasets, i.e., put them as 2 separate sections.
(6) For "Section 3.3: Correlations between CO2 emissions and human pressures", please include some insights obtained based on these statistical figures, for example, from political perspectives / government planning aspects?
Other than all aforementioned changes needed, I think the manuscript is of scientific impact and is quite meaningful.
Author Response
Special thanks to you for your good comments. Detailed comments and our detailed responses/corrections are listed as below:
- Referring to point (2): About the appropriateness and technical deficiencies of the use of remotely sensed datasets, the authors should at least mention this in a humble manner, either at the beginning of Sec. 2, or at the end of this manuscript. They should include the response contents into the manuscript somehow - i.e., mentioning that government-based datasets have certain advantages in terms of quantitative analysis, and the next goal is to prepare a bottom-up carbon footprint and conduct corresponding analyses - some future goals have to be explicitly mentioned.
Response: Thanks for your comment. We have added the current deficiencies and future prospects are in the conclusion on Line 426-432, Page 9 in the revised manuscript.
“The remote sensing technology has expanded human’s ability to understand their living environment, and has the advantages of qualitative accuracy, large observation range, high spatial resolution, simple acquisition, and strong data consistency. However, government-based datasets have certain advantages in terms of quantitative analysis. Our next goal is to conduct a bottom-up carbon footprint coupled with remote sensing data and government-based datasets for the analysis of the impact of human activities on car-bon emissions.”
- Referring to point (2): Section 2.2 - Section 2.7 are all related to datasets adopted in this study. Some references should be added to explain how these types / categories of datasets were being adopted in the past for similar analyses / spatial analyses. This could enhance the confidence of usage of remotely sensed datasets. For example,
Response: Thanks for your comment and suggestion. We have added some references to explain HF model.
References:
- Din, S. U., et al. Retrieval of land-use/land cover change (LUCC) maps and urban expansion dynamics of Hyderabad, Pakistan via Landsat Datasets and support vector machine framework. Remote Sens. 13, 3337 (2021).
- Arshad, S. et al. Change detection of land cover/land use dynamics in arid region of Bahawalpur District, Pakistan. Appl. Geomat. 14, 387–403 (2022).
- Yeboah, G., et al. Analysis of Openstreetmap data quality at different stages of a participatory mapping process: evidence from Slums in Africa and Asia. ISPRS Int. J. of Geo-Inf. 10, 265 (2021).
- Wang, X. Y., et al. Projecting 1 km-grid population distributions from 2020 to 2100 globally under shared socioeconomic pathways. Sci. Data 9, 563 (2022).
- Gaughan, A. E. et al. Evaluating nighttime lights and population distribution as proxies for mapping anthropogenic CO2 emission in Vietnam, Cambodia and Laos. Environ. Res. Commun. 1, 091006 (2019).
- Shi, K. F. et al. Carbon dioxide (CO2) emissions from the service industry, traffic, and secondary industry as revealed by the remotely sensed nighttime light data. Int. J. Digit. Earth, 14, 1514–1527 (2021).
- Zhang, Y. F. et al. Spatial variations in CO2 fluxes in a subtropical coastal reservoir of Southeast China were related to urbanization and land-use types. J. Environ. Sci. 109, 206–218, (2021).
- Referring to point (4): HF has now been explained, which sounds great, but some details of past studies between HF and CO2 emission figures should be provided as well.
Response: Thanks for your comment. We have added some details of past studies between HF and CO2 emission on Line 114-117, Page 3 in the revised manuscript. At present, there are few studies on the relationship between human footprint and carbon emissions. Human footprint can reflect human disturbance to natural environment, and is directly or indirectly related to human fossil fuel combustion, fertilizer use, and industrial activity, thus establishing a certain relationship with carbo CO2 emissions.
References:
Doney S. C. The Growing Human Footprint on Coastal and Open-Ocean Biogeochemistry. Science 5985, 1512–1516 (2010).
- In "Section 1: Introduction", please include the scientific advancement and breakthrough made in current manuscript in an explicit manner. Further, please include a short paragraph describing the main role / idea of each Section.
Response: Thanks for your comment. We have added the scientific advancement, breakthrough, and a short paragraph describing the main idea of each Section in the last paragraph of Introduction on Line 101-111, Page 3 in the revised manuscript.
“Our results enhance the understanding of the impacts of human pressures on the environment and have implications for the formulation of effective and environmentally friendly strategies. The results have a positive guiding significance for carbon emission reduction policies, including urban land planning, population size control, industrial structure optimization, and industrial technology upgrading, etc. In this paper, Section 2 provides a brief overview of the study methods, data sources, and model construction. Then, the spatio-temporal characteristics of HF and CO2, and their correlation analysis during 2000–2017 in China are provided in Section 3. The cause of spatial distribution of HF and CO2, and response and driving factors of CO2 emissions to human pressure changes are discussed in Section 4. Finally, Section 5 outlines the potential extension, out-look, and the summary of this study.”
- For "Section 2: Methods", I think it should be "2. Methods and Data", or the authors should separate Methods from Datasets, i.e., put them as 2 separate sections
Response: Thanks for your comment. We have revised the subtitle “Methods and Data” in the revised manuscript.
- For "Section 3.3: Correlations between CO2 emissions and human pressures", please include some insights obtained based on these statistical figures, for example, from political perspectives / government planning aspects?
Response: Thanks for your comment. We have mainly summarized three parts of opinions and suggestions on CO2 emission reduction for explaining “the correlations between CO2 emissions and human pressures”: 1) technology transfer and optimization of the industrial structure; 2) controlling the rate of increase in human pressures including human population, land-use structure, and urban construction; 3) developing UHV and NEVs technology in Section 4.3 on Line 367-375, Page 8; Line 385-390, Page 8; Line 401-407, Page 9 in the revised manuscript.
“Heavy industries are central to the economies of the northern provinces, and these products are mainly exported to other provinces. Such provinces do not have adequate hu-man resources to upgrade their technologies and equipment, and the economies of these provinces are mainly based on energy-intensive industries, such as mining, metals, electricity, and chemical products; consequently, the resource use efficiency of these provinces is low, and their greenhouse gas emissions are high. China’s economy is largely de-pending on primary energy resources, and these resources are mainly located in less developed regions. Technology transfer and optimization of the industrial structure are important for reducing CO2 emissions in less developed regions.”
“Modifications of the energy structure would be an effective approach for reducing CO2 emissions in these regions. Controlling the rate of increase in human pressures can also contribute to mitigating CO2 emissions; there is thus a need to reduce the effect of human pressures on the environment, especially the deleterious effects of the human population, land-use structure, and urban construction.”
“China has been committed to developing UHV technology for many years. As of 2019, more than 20 UHV transmission lines have been built in China. In addition, the Chinese government is vigorously promoting the use of new energy vehicles (NEVs). Annual sales of NEVs were 1.367 million in 2020, which is more than 160 times the number of NEVs sold in 2011 (8,159). Direct policy support has promoted the rapid development of China’s NEV market and has made large contributions to energy saving and emission reductions.”

Reviewer 2 Report
I think the revised manuscript can be acceptable
Author Response
Special thanks to you for your good comments. Your proposal has been greatly improved for our manuscript.